# Size-aware Compression of 3D Gaussians with Fine-grained Mixed Precision Quantization

## Abstract

In this paper, we propose a method to automatically select hyperparameters to compress 3D Gaussians to a target file size while maximizing visual quality. We iteratively search for a hyperparameter configuration until the file size meets the specified budget. However, existing compression frameworks require completing the entire compression process to determine the compressed file size, which is time-consuming. To accelerate this, we design a tailored size estimator for frameworks that can determine hyperparameters without requiring fine-tuning. Although the finetuning-free frameworks are more predictable, they typically underperform compared to fine-tuning-based approaches, which utilize end-to-end differentiable structures to achieve superior results. To close this performance gap, we propose a mixed-precision quantization strategy that exploits the heterogeneity of attribute channels by compressing each channel with different bit-widths. The resulting combinatorial optimization problem is efficiently solved using 0-1 integer linear programming. Additionally, we partition each attribute channel into blocks of vectors, quantizing each vector based on the optimal bit-width determined in the previous step. The block length is then determined via dynamic programming. Our method identifies hyperparameter settings that meet the target file size within 70 seconds, outperforming state-of-the-art methods in both efficiency and quality. Extensive experiments demonstrate that our approach significantly enhances the performance of fine-tuning-free methods, with its upper-bound performance comparable to that of fine-tuning-required techniques.

## 1 Introduction

In recent years, 3D Gaussian Splatting (3DGS) (Kerbl et al., 2023) has emerged as a popular research topic due to its excellent quality and real-time rendering speed in novel view synthesis. 3DGS represents a 3D scene using a set of neural Gaussians initiated from Structure-from-Motion (SfM) with learnable attributes such as color, shape, and opacity. The 2D images can be effectively rendered using differentiable rasterization and end-to-end training is enabled. Meanwhile, benefiting from efficient CUDA implementation, real-time rendering is achieved. Despite its success, 3DGS still encounters limitations in storage efficiency. For example, $5.27 10^6$ Gaussians are required to represent the bicycle scene in the Mip-NeRF 360 dataset (Barron et al., 2022), occupying 1.3 GB of storage under 32-bit float precision. This sizable file poses challenges in transmission and storage. Hence, a tailored codec for 3D Gaussians is required.

Current 3D Gaussian compression frameworks can be categorized into two types: finetuning-free (Xie et al., 2024; Niedermayr et al., 2024; Fan et al., 2023; Papantonakis et al., 2024) and finetuning-required (Lee et al., 2024; Chen et al., 2024; Morgenstern et al., 2024; Papantonakis et al., 2024; Wang et al., 2024). The finetuning-free approach allows direct compression of a fully-trained 3DGS using given hyperparameters, without any additional training. In contrast, the finetuning-required approach involves training a module from scratch as part of the compression process, such as learning a pruning mask (Lee et al., 2024) or a context model (Chen et al., 2024). To compress the 3DGS to meet a specific size budget while achieving optimal visual quality, we have to iteratively search for a hyperparameter configuration until the file size meets the size budget. However, neither of these frameworks cannot support such an operation because they require completing the entire compression process to determine the final file size. For example, finetuning-free methods

like MesonGS typically take about one minute to complete compression, while finetuning-required methods can take up to 20 minutes, as they need to learn a pruning mask from scratch to remove Gaussian points. In addition, the file size fluctuates during the finetuning process, making the final size harder to predict. As a result, we chose MesonGS (Xie et al., 2024), a finetuning-free method, as the foundation of our approach and designed a tailored size estimator for it.

As ScaffoldGS (Lu et al., 2024) became a mainstream basemodel for 3D Gaussian compression, we first migrate MesonGS to ScaffoldGS. Then, we design a size estimator tailored to the ScaffoldGS-based MesonGS, which can predict the final size with negligible latency under a given hyperparameter setting. Based on the size estimator and the observation of heterogeneity across channels of attributes, we formulate the size-aware mixed-precision quantization as an Integer Linear Programming (ILP) problem to find the best bit-precision setting. Compared to the contemporary work (Chen et al., 2024) - HAC, our approach leverages the differences between different channels of attributes, while HAC cheats the different channels of attributes as the same. Furthermore, given the number of blocks, we have to slice each channel of attributes into multiple blocks, we propose a dynamic programming to determine the length of each block. Our main contributions can be listed as follows:

- We propose an estimator to predict the final size of compressed 3D Gaussians before compression. This Size Estimator can help quickly search for hyperparameter settings that meet the size budget while maximizing visual quality.

- Based on the estimator, we propose a size-aware hierarchical mixed precision quantization scheme. On the inter-attribute level, we formulate mixed bit-width selection as a 0-1 ILP problem subject to the size. On the intra-attribute level, we propose a high-speed dynamic programming algorithm to solve the mixed block-length setting inside a channel.

- Our method can find hyperparameter settings that meet the size budget and maximize visual quality within 70 seconds, which is $100\times$ faster than baselines. Furthermore, our mixed-precision method addresses the shortcomings of fine-tuning-free approaches, achieving performance comparable to or even better than state-of-the-art results across multiple datasets.

## 2 RELATED WORK

### 2.1 3D GAUSSIAN SPLATTING

3D Gaussian Splatting (Kerbl et al., 2023) is a novel method for reconstructing 3D scenes from 2D images. It represents a scene using a set of 3D Gaussian distributions, which are small, fuzzy blobs. These Gaussians capture the density, color, and opacity of the scene. It is faster and less memory-intensive than previous methods like NeRF (Mildenhall et al., 2021) and can produce high-quality, smooth results with fewer artifacts. Recently, a lot of work has been proposed for compressing 3D Gaussians. At first, they focused on compressing 3DGS model (Niedermayr et al., 2024; Lee et al., 2024; Girish et al., 2024; Morgenstern et al., 2024; Navaneet et al., 2024; Papantonakis et al., 2024; Fan et al., 2023; Xie et al., 2024). But now, many works pay attention to compressing a more efficient GS model (Fang & Wang, 2024; Lu et al., 2024), in which ScaffoldGS (Lu et al., 2024) became the hot spot. It proposed to divide anchors into voxels and introduce an anchor feature for each voxel to grasp the common attributes of neural Gaussians in the voxel, i.e., the neural Gaussians are predicted by the anchor features. HAC (Chen et al., 2024) recognized the advantages of ScaffoldGS and proposed a tailored compression method for it, extracting a context from the 3D coordinates to guide the quantization steps and entropy encoding parameters. ContextGS (Wang et al., 2024) divides anchors into hierarchical levels and encodes them progressively. However, existing compression frameworks for ScaffoldGS often require training from scratch, and thus the size varies greatly over time, making size estimation under given hyperparameters challenging. This paper transfers the relatively easy-to-estimate MesonGS to ScaffoldGS and conducts an in-depth analysis of the compression components of MesonGS to build a nearly delay-free size estimator.

### 2.2 MIXED-PRECISION QUANTIZATION

Significant efforts have recently been made to improve the trade-off between the accuracy and efficiency of neural networks. A promising direction is to use mixed-precision quantization (Wang et al., 2019; Dong et al., 2019; 2020; Yao et al., 2021; Tang et al., 2022). However, the challenge

with this approach is to find the right mixed-precision setting for the different layers of neural networks. A brute force approach is not feasible since the search space is exponentially large in the number of layers. HAQ (Wang et al., 2019) proposes to employ reinforcement learning (RL) to search this space. However, this RL-based solution requires tremendous computational resources. HAWQ (Dong et al., 2019; 2020; Yao et al., 2021) proposes first to assign each layer a sensitivity score with the Hessian spectrum and then formulate an ILP solution that can generate mixed-precision settings with various constraints (such as model size, BOPS, and latency). Unlike the challenges faced by neural networks, all attributes of 3DGS are equivalent to the weights in neural networks, and two levels of mixed precision need to be selected: inter-attributes and intra-attributes. Moreover, 3DGS demands a higher compression speed than neural networks. We identify an opportunity to apply mixed precision quantization to attributes of the GS model and propose a hierarchical scheme to quickly determine the optimal mixed-precision setting for the attribute. Besides, 3DGS compression has 16 bit options available, far exceeding 2 bit options of HAWQ (INT4 and INT8). Directly using the integer programming formulation from HAWQ3 makes it difficult to achieve good results. To address this, we establish a more general and fast 0-1 integer programming formulation to determine the optimal bit-width for each attribute channel.

## 3 PRELIMINARY

**3D-GS** (Kerbl et al., 2023) is an explicit 3D scene representation in the form of point clouds, utilizing Gaussians to model the scene. Each Gaussian is characterized by a covariance matrix $\Sigma$ and a center point $\mu$, which is referred to as the mean value of the Gaussian: $G(x) = e^{-\frac{1}{2}(x-\mu)^\top \Sigma^{-1}(x-\mu)}$. To maintain the positive definiteness of the covariance matrix $\Sigma$, 3D-GS decomposes $\Sigma$ into a scaling matrix $\mathbf{S} = \text{diag}(\mathbf{s}), \mathbf{s} \in \mathbb{R}^3$ and a rotation matrix $\mathbf{R}$: $\Sigma = \mathbf{RSS}^\top \mathbf{R}^\top$. The rotation matrix $\mathbf{R}$ is parameterized by a rotation quaternion $\mathbf{q} \in \mathbb{R}^4$. The backpropagation process is illustrated in (Kerbl et al., 2023).

When rendering novel views, the technique of splatting (Zwicker et al., 2001a; Yifan et al., 2019) is employed for the Gaussians within the camera planes. As introduced by (Zwicker et al., 2001b), using a viewing transform denoted as $\mathbf{W}$ and the Jacobian $\mathbf{J}$ of the affine approximation of the projective transformation, the covariance matrix $\Sigma'$ in camera coordinates system can be computed by $\Sigma' = \mathbf{JW\Sigma W}^\top \mathbf{J}^\top$.

In summary, each element of 3D Gaussians has the following parameters: (1) a 3D center $\mu \in \mathbb{R}^3$; (2) a rotation quaternion $\mathbf{q} \in \mathbb{R}^4$; (3) a scale vector $\mathbf{s} \in \mathbb{R}^3$; (4) a color feature defined by spherical harmonics coefficients $\mathbf{SH} \in \mathbb{R}^h$, with $h = 3(d+1)^2$, where $d$ is the harmonics degree; and (5) an opacity logit $o \in \mathbb{R}$. Specifically, for each pixel, the color and opacity of Gaussians are computed using $G(x)$. The blending of $N$ ordered points that overlap the pixel is given by: $C = \sum_{i \in N} c_i \alpha_i \prod_{j=1}^{i-1}(1 - \alpha_j)$. Here, $c_i$ and $\alpha_i$ represent the density and color of this point computed by a Gaussian with covariance $\Sigma$ multiplied by a per-point opacity and SH color coefficients.

**Scaffold-GS** (Lu et al., 2024) is a variant of 3DGS, widely adopted in 3DGS compression due to its low storage requirements. It introduces *anchor points* to capture common attributes of local 3D Gaussians. Specifically, the *anchor points* are initialized from neural Gaussians by voxelizing the 3D scenes. Each anchor point has a context feature $\mathbf{f} \in \mathbf{R}^{32}$, a location $\mathbf{x} \in \mathbf{R}^3$, a scaling factor $\mathbf{l} \in \mathbf{R}^6$ and $k$ learnable offset $\mathbf{O} \in \mathbf{R}^{k \times 3}$. Given a camera at $\mathbf{x}_c$, anchor points are used to predict the view-dependent neural Gaussians in their corresponding voxels as follows,

$$\{\mathbf{c}^i, \mathbf{r}^i, \mathbf{s}^i, \alpha^i\}_{i=0}^k = F(\mathbf{f}, \boldsymbol{\sigma}_c, \vec{\mathbf{d}}_c), \tag{1}$$

where $\boldsymbol{\sigma}_c = ||\mathbf{x} - \mathbf{x}_c||_2$, $\vec{\mathbf{d}}_c = \frac{\mathbf{x} - \mathbf{x}_c}{||\mathbf{x} - \mathbf{x}_c||_2}$, the superscript $i$ represents the index of neural Gaussian in the voxel, $\mathbf{s}^i, \mathbf{c}^i \in \mathbf{R}^3$ are the scaling and color respectively, and $\mathbf{r}^i \in \mathbf{R}^4$ is the quaternion for rotation. In the left side of the Fig. 1, the positions of neural Gaussians are then calculated as

$$\{\boldsymbol{\mu}^0, ..., \boldsymbol{\mu}^{k-1}\} = \mathbf{x} + \{\mathbf{O}^0, ..., \mathbf{O}^{k-1}\} \cdot \mathbf{l}_{:3}, \tag{2}$$

where $\mathbf{x}$ is the learnable positions of the anchor and $\mathbf{l}_{:3}$ is the base scaling of its associated neural Gaussians. After decoding the properties of neural Gaussians from anchor points, other processes are the same as the 3DGS (Kerbl et al., 2023). By predicting the properties of neural Gaussians from the anchor features and saving the properties of anchor points only, Scaffold-GS greatly eliminates the redundancy among 3D neural Gaussians and decreases the storage demand.

Figure 1: **Overview.** The left side shows the process of generating 3D Gaussian splats from the anchor points, along with the data associated with them, including features, scaling, offsets, and location. Then, from the middle to the right, the figure illustrates how the MesonGS compression framework is applied to ScaffoldGS representation. The specific process involves pruning unimportant anchor points, performing voxelization on the locations to generate an octree, applying Region Adaptive Hierarchical Transform (RAHT) to the attributes based on the voxelized locations to generate DC and AC coefficients, then quantizing the AC coefficients using hierarchical mixed-precision quantization, and finally packing all the elements. The design of the Size Estimator is based on the MesonGS framework, which first determines the pruning ratio $\tau$, octree depth $d$, and number of blocks $n_b$. Subsequently, the size estimator can also help solve the bit-width settings for each channel of attributes, given a specified size budget (Sec. 4.2). Finally, we use dynamic programming to further optimize the block partitioning for each channel, enhancing the compression within the given size budget.

## 4 METHODOLOGY

### 4.1 SIZE ESTIMATOR FOR COMPRESSED 3D GAUSSIAN FILE

Estimating the file size for compressed 3DGS is nontrivial. First, the difficulty of size estimation varies across different frameworks. We observe that the more modules that require training, the harder it becomes to estimate the final file size. The learnable masking module proposed by (Lee et al., 2024) requires training; in HAC (Chen et al., 2024), context information extracted from the coordinates (represented using a Hash Grid) needs to be trained from scratch, as well as the hyperparameters used for entropy coding. These representations, which need to be learned from scratch, result in a compressed file size that is difficult to predict. In contrast, frameworks like MesonGS, which allow for offline compression, are easier to predict since none of the modules require training. As shown in Fig. 2a, the final file size of HAC fluctuates continuously during training, while the size of MesonGS remains very stable with almost no variation. Therefore, we adopt the MesonGS framework for compression.

Since ScaffoldGS is currently the base model of the state-of-the-art 3D Gaussian compression works (Chen et al., 2024; Wang et al., 2024; Bagdasarian et al., 2024), we also adopt it as the base model for our approach. Hence, we migrate MesonGS onto ScaffoldGS and the compression process can be summarized as follows. As illustrated in the middle of Fig. 2, we first calculate the importance of each anchor point by averaging the importance of Gaussian splats generated by it. Then, we prune the anchor points based on a hyperparameter pruning percentage $\tau$. The coordinates of the anchor points are voxelized to form an Octree. For anchor points within the same voxel, we average the attributes to ensure that each voxel corresponds to only one anchor point. We denote the number of points that after pruning and voxelization as $N$. Based on the voxelized coordinates, we apply region-adaptive hierarchical transform (RAHT) to all attributes, which produces the DC and AC coefficients. DC coefficients are stored in float format while the AC coefficients are then quantized in a block-wise manner. We denote the channels of attributes as $M$. Here, block-wise quantization means dividing each channel's attributes into $K$ blocks and then quantizing each block into the bit width of 8. Here, elements in a block share the same scale and zero-point parameters. Finally, all components are compressed by LZ77 (Gailly & Adler, 2003; Ziv & Lempel, 1977; 1978) codec as shown in the right side of the Fig. 1. The metadata includes the octree depth $d$ and the number of blocks $K$.

By analyzing the above compression process, we identify the hyperparameters that impact the final model size are pruning percentage $\tau$, octree depth $d$, and the number of blocks $K$. First, to simplify

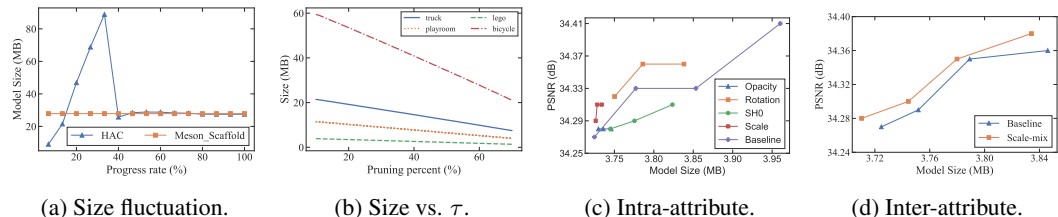

| (a) Size fluctuation. | (b) Size vs. $\tau$. | (c) Intra-attribute. | (d) Inter-attribute. |

Figure 2: Motivation of size estimation and mixed precision quantization.

the influence of voxelization on the final size, we aim to provide the finest possible subdivision. This ensures that the octree depth is sufficient, so the number of anchor points is minimally reduced after voxelization. Next, we consider the effect of the pruning percentage on the final size. As shown in Fig. 2b, we observe a linear relationship between the pruning percentage and the final file size. Finally, as MesonGS quantizes all of the blocks into the same bit width, we can estimate the bit-width of the compressed AC coefficient as $(N-1) * H * 8$. In summary, given a fixed pruning percentage $\tau$, the final file size $S$ can be estimated using the following equation:

$$S = \frac{1}{8}(\underbrace{(N-1)*H*8}_{\text{AC}} + \underbrace{H*32}_{\text{DC}} + \underbrace{N*64}_{\text{Octree}} + \underbrace{H*K*32*2}_{\text{Scales \& Zero points}} + \underbrace{2*32}_{\text{depth}, K}) + \underbrace{50*1024}_{\text{MLP}}. \quad (3)$$

This size estimator formula is built for a universal bit-width setting. We will update this formula in Sec. 4.2 and Sec. 4.3 to ensure that the final size prediction is accurate.

### 4.2 INTER-ATTRIBUTE MIXED PRECISION QUANTIZATION

Although post-training compression techniques like MesonGS are more convenient for estimating the final size, pre-fixing hyperparameters hinder the model from achieving optimal performance. The core reason lies in the fact that MesonGS uses a uniform quantization precision, which prevents it from obtaining better compression results. As shown in Fig. 2, we observe that different attributes are better suited to varying levels of compression granularity. Moreover, a block strategy with mixed granularity yields better results for the same attribute. Besides, *attributes occupy over 90% of the final storage, further compressing them offers the greatest potential benefit.*

Uniformly quantizing all the attributes to low-bitwidth (e.g. INT4) could lead to significant quality degradation. However, it is possible to benefit from low-precision quantization by keeping a subset of sensitive attributes at high precision. The basic idea is to keep sensitive channels at higher precision and insensitive layers at lower precision. An important component of MPGS is that we directly consider the *size* metric, to select the bit-precision configuration. Previous methods have been unable to control the file size of the GS model. For example, given a desired file size, it has been difficult to identify the bit configuration that is closest to this size while achieving the optimal quality. In this work, we formalize the problem as an Integer Linear Programming (ILP) problem.

Assume that for each attribute channel, there are $B$ quantization options (e.g., 2 options for INT4 or INT8). For the AC coefficients with $H$ attribute channels, the search space of the ILP is $B^H$. The objective of solving the ILP is to find the best bit configuration among these $B^H$ possibilities that optimally balances information loss $\Omega$ and the user-specified file size. Each bit-precision setting can lead to a different model perturbation. To simplify the problem, we assume that the perturbations of each channel are independent of one another. This allows us to precompute the information loss of each channel separately, and it only requires $BH$ computations. For the information loss metric, we use the mean square error between the original attributes and the restored attributes[1]. Formally, we can precompute the information loss matrix $\Omega \in \mathbb{R}^{H \times B}$ with:

$$\Omega(i, j) = \|\hat{\mathcal{A}}_i^j - \mathcal{A}_i\|^2. \quad (4)$$

---

[1]Similar assumption can be found in (Dong et al., 2019; 2020).

The ILP problem tries to find the right bit precision $\mathbf{Q} \in \{0,1\}^{H \times B}$ that minimizes the information loss, as follows:

$$\text{Objective: } \Omega \odot \mathbf{Q}, \tag{5}$$

$$\text{Subject to: } S(\mathbf{Q}) \leq \text{ Model Size Limit}, \tag{6}$$

$$\forall i \in \{0, ..., H-1\}, \sum_{j=0}^{B} \mathbf{Q}_{i,j} = 1. \tag{7}$$

Here, $S(\cdot)$ denotes the compressed file size of the GS model under the bit-configuration of $\mathbf{Q}$. Here, to adapt the size estimator to mixed bit-widths, we replace the size estimation term of AC $8(N-1)H$ with $(N-1)\mathbf{w}\mathbf{Q}$ in Eq. 3, where $\mathbf{w}$ is $[1, 2, ..., B]$. We set $B$ as 16. Note that we cannot solve out the bit-width setting if there is no size estimator. Since the number of blocks and the bit-width setting influence each other, we fix the number of blocks and only adjust the bit-width.

We solve this 0-1 ILP using open source PULP library (Roy & Mitchell, 2020) in Python, where we found that for all the configurations tested in the paper, the solver can find the solution in less than 1 second given the sensitivity metric.

### 4.3 Intra-Attribute Mixed Precision Quantization

Given a desired model size limitation, the ILP solver in Sec. 4.2 generates optimal intra-attribute bit precision configurations for different channels of attributes. In this part, we develop methods at the single attribute level to optimally slice a channel of attribute into $n_b$ blocks. Our optimization goal is to minimize the permutation loss for each channel of the attribute when slicing a channel of attributes into $n_s$ blocks. Note that the size of each block does not have to be the same.

Assume that we want to split a channel of attributes with length $N$ into $K$ blocks, noted as $\{b_1, b_2, ..., b_K\}$, and then quantize each block to $q$ bits. We denote the start index of each block as $\{n_1, n_2, ..., n_K\}$. Our goal is to minimize the information loss caused by block-wise quantization. Following the metrics proposed by mixed precision quantization for deep learning models (Dong et al., 2019; 2020; Yao et al., 2021), the minimal information loss for quantizing this channel of attribute is written as:

$$L^* = \min_{n_1, n_2, ..., n_K} \left\{ \sum_i L(n_i, n_{i+1}) \right\}, \quad L(n_i, n_{i+1}) = \frac{\|\hat{b}_i - b_i\|^2}{n_{i+1} - n_i}, \tag{8}$$

where $\hat{b}_i$ refers to the vector that dequantized from the quantized $b_i$. We measure the information loss of quantizing a block vector as the mean square error between $\hat{b}_i$ and $b_i$.

**DP formulation.** To find $L^*$, we develop a DP algorithm. Specifically, we use the function $F(k, l)$ to represent the minimal total information loss when slicing the $l$ elements into $k$ blocks. We start with $F(0, 0) = 0$, and derive the optimal substructure of $F$ as follows:

$$F(k, l) = \min_{0 < i \leq l-k} \{L_{l-i, l} + F(k-1, l-i)\}. \tag{9}$$

**Complexity.** Our DP algorithm first iterates over all possible $k$ and $l$. Then, for each $F(k, l)$, the DP algorithm traverses through $l - k$ combinations to select the one with the minimum loss. For each combination, we iterate through blocks of length $i$ to compute the information loss. The overall complexity is $\mathcal{O}(KN(N-K)^2)$. Since $K \ll N$, the final complexity becomes $\mathcal{O}(KN^3)$. Here, $N$ represents the length of the attribute, which is also the number of points in the GS model. Hence, this complexity is not feasible in practice because the number of points in a typical ScaffoldGS model is around $80,000$. To accelerate this DP process, we limit the step size for each DP iteration to multiples of $U$, reducing the complexity to $\mathcal{O}(K(N/U)^3)$.

Note that incorporating mixed block lengths does not significantly affect the final file size. For mixed block length quantization, we only need to record the starting index of each block to ensure decompression. The size of these indices is generally $32NH$.

Besides, although HAC employs mixed-precision quantization, its granularity is relatively coarse. The reason is that the quantization granularity of HAC is cross-channel, whereas our method employs a finer-grained, intra-channel quantization. More illustrations are provided in the Appendix.

Table 1: Latency and quality of searched results, under a file size budget.

| Method | Target Size (B) | PNSR | SSIM | LPIPS | Searched Size (B) | Search Time (s) |
|--------|-----------------|-------|--------|--------|-------------------|-----------------|
| HAC | 30,000,000 | 25.07 | 0.7424 | 0.2633 | 29,999,969 | 7725.32 |
| Our | | 25.16 | 0.7446 | 0.2606 | 29,946,299 | 67.42 |

### 4.4 SEARCHING AND FINETUNING

Given a target file size, we first randomly select two $\tau$ values and perform the compression process under 8-bit precision to obtain the final file sizes. Then, we substitute these results into the equation $size = \alpha\tau + \beta$ to solve for the parameters $\alpha$ and $\beta$, as illustrated in Fig 2b. Afterward, we can solve for the $\tau$ corresponding to the target file size. Finally, we employ ILP and DP to determine the bit-width and block length settings.

During fine-tuning, we fix the pruning mask and the 3D coordinate. We noticed that the fine-tuning speed of the MesonGS is relatively slow and found this is due to MesonGS applying quantization block by block. To accelerate this process, we implemented a CUDA kernel to parallelly run the block-wise quantization.

## 5 EXPERIMENTS

**Datasets.** We conduct experiments on four datasets: 1) **Mip-NeRF 360 (Barron et al., 2022).** This dataset contains five outdoor and four indoor scenes. Each scene contains 100 to 300 images. We use the images at 1600×1063. 2) **Tank & Temples (Knapitsch et al., 2017).** This dataset contains two scenes, including *train* and *truck*. 3) **Deep Blending (Hedman et al., 2018).** This dataset contains two scenes, including *drjohnson* and *playroom*. 4) **Synthetic-NeRF (Mildenhall et al., 2021).** This is a view synthesis dataset consisting of 8 synthetic scans, with 100 views used for training and 200 views for testing.

**Baselines.** We compare our method with the following baselines: 3DGS (Kerbl et al., 2023), Scaf-foldGS (Lu et al., 2024), C3DGS (Niedermayr et al., 2024), Lee *et al.* (Lee et al., 2024), Light-Gaussian Fan et al. (2023), EAGLES (Girish et al., 2024), SOGS (Morgenstern et al., 2024), Com-pact3D (Navaneet et al., 2024), ReduGS Papantonakis et al. (2024), MesonGS (Xie et al., 2024), HAC (Chen et al., 2024), DVGO (Sun et al., 2022), VQRF (Li et al., 2023), and ACRF (Fang et al., 2024). Quantitative results of baseline methods are derived from HAC (Chen et al., 2024) and MesonGS (Xie et al., 2024), while the qualitative results are produced from our experiments.

### 5.1 EXPERIMENTAL RESULTS

**End-to-end Performance.** Our method is proposed to solve the problem of automatically selecting hyperparameters to compress 3D Gaussians under a size budget while maximizing visual quality. We evaluate this ability of our method via latency and quality metrics. In Tab. 1, our method is $100\times$ faster than HAC and achieves better compression quality. Here, the latency of the HAC baseline is measured by tuning the hyper-parameter $\lambda_e$. We tried $\lambda_e$ from 0.04 to 0.037, 4 times, to find a suitable file size of HAC.

**Quantitative Comparison.** The quantitative compression results of different methods are presented in Tab. 2 and Fig. 4. Our method outperforms most others across all three datasets and achieves performance comparable to the SOTA method – HAC, demonstrating that MPQ effectively addresses the shortcomings of post-training compression methods. We also provide a comparison with NeRF compression, as shown in Tab. 3. Our method achieves better performance with a smaller file size.

**Qualitative Comparison.** The qualitative results are depicted in Fig. 3. We present the render-ing results and the corresponding error maps. From the error maps, it is evident that our method handles chair reflections better than other methods while achieving rendering results comparable to ScaffoldGS.

**Encoding and Decoding Time Comparison.** We also compare the encoding and decoding time of the ScaffoldGS-based approach in Tab. 4. For consistency and fairness across all experiments, we utilize a virtual machine equipped with 14 vCPU of Intel(R) Xeon(R) Platinum 8362 CPU @

Table 2: Quantitative results of our approach and others. The best and 2nd best results are highlighted in red and yellow cells. Note that we do not consider 3DGS and ScaffoldGS when highlighting this. The size is measured in MB.

| Method | Mip-NeRF 360 | | | | Tank&Temples | | | | Deep Blending | | | |
|---|---|---|---|---|---|---|---|---|---|---|---|---|
| | PSNR | SSIM | LPIPS | Size | PSNR | SSIM | LPIPS | Size | PSNR | SSIM | LPIPS | Size |
| 3D-GS | 27.49 | 0.813 | 0.222 | 744.7 | 23.69 | 0.844 | 0.178 | 431.0 | 29.42 | 0.899 | 0.247 | 663.9 |
| ScaffoldGS | 27.50 | 0.806 | 0.252 | 253.9 | 23.96 | 0.853 | 0.177 | 86.50 | 30.21 | 0.906 | 0.254 | 66.00 |
| Lee *et al.* | 27.08 | 0.798 | 0.247 | 48.80 | 23.32 | 0.831 | 0.201 | 39.43 | 29.79 | 0.901 | 0.258 | 43.21 |
| C3DGS | 26.98 | 0.801 | 0.238 | 28.80 | 23.32 | 0.832 | 0.194 | 17.28 | 29.38 | 0.898 | 0.253 | 25.30 |
| EAGLES | 27.15 | 0.808 | 0.238 | 68.89 | 23.41 | 0.840 | 0.200 | 34.00 | 29.91 | 0.910 | 0.250 | 62.00 |
| LightGaussian | 27.00 | 0.799 | 0.249 | 44.54 | 22.83 | 0.822 | 0.242 | 22.43 | 27.01 | 0.872 | 0.308 | 33.94 |
| SOGS | 26.01 | 0.772 | 0.259 | 23.90 | 22.78 | 0.817 | 0.211 | 13.05 | 28.92 | 0.891 | 0.276 | 8.40 |
| Compact3d | 27.16 | 0.808 | 0.228 | 50.30 | 23.47 | 0.840 | 0.188 | 27.97 | 29.75 | 0.903 | 0.247 | 42.77 |
| ReduGS | 27.10 | 0.809 | 0.226 | 29.00 | 23.57 | 0.840 | 0.188 | 14.00 | 29.63 | 0.902 | 0.249 | 18.00 |
| MesonGS | 26.98 | 0.801 | 0.233 | 28.77 | 23.32 | 0.837 | 0.193 | 16.99 | 29.51 | 0.901 | 0.251 | 24.76 |
| HAC | 27.53 | 0.807 | 0.238 | 15.26 | 24.04 | 0.846 | 0.187 | 8.10 | 29.98 | 0.902 | 0.269 | 4.35 |
| Our-big | 27.65 | 0.809 | 0.235 | 21.63 | 24.01 | 0.835 | 0.200 | 11.99 | 30.25 | 0.904 | 0.271 | 5.99 |
| Our-middle | 27.45 | 0.806 | 0.241 | 17.62 | 23.94 | 0.835 | 0.198 | 10.24 | 30.14 | 0.901 | 0.276 | 7.42 |
| Our-small | 27.21 | 0.799 | 0.251 | 13.54 | 23.80 | 0.834 | 0.202 | 8.28 | 29.82 | 0.896 | 0.287 | 4.53 |

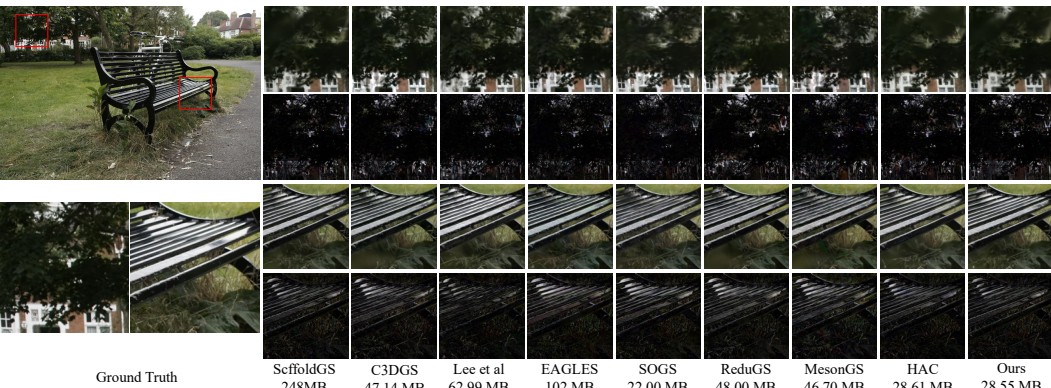

| Ground Truth | ScffoldGS 248MB | C3DGS 47.14 MB | Lee et al 62.99 MB | EAGLES 102 MB | SOGS 22.00 MB | ReduGS 48.00 MB | MesonGS 46.70 MB | HAC 28.61 MB | Ours 28.55 MB |

Figure 3: Qualitative results (*bicycle*) of our method and other baselines. We present the rendering results (rows 1 and 3) along with the corresponding error maps (rows 2 and 4).

2.80GHz and an NVIDIA 3090 RTX GPU. The encoding time here refers to the training time per iteration. The decoding speed of our method is faster. In addition, our CUDA implementation boosts the speed by $2.69\times$.

Table 3: Quantitative comparison on Synthetic-NeRF.

| Method | PNSR | SSIM | LPIPS | Size (MB) |
|---|---|---|---|---|
| DVGO | 31.90 | 0.956 | 0.035 | 105.92 |
| VQRF | 31.77 | 0.954 | 0.036 | 1.43 |
| ACRF | 31.79 | 0.954 | 0.037 | 1.15 |
| Our | 32.41 | 0.960 | 0.043 | 1.10 |

Table 4: We compare the encoding (*Enc*) and decoding (*Dec*) times with HAC and MesonGS.

| Method | Enc (s) | Dec (s) |
|---|---|---|
| HAC | 0.07 | 8.78 |
| MesonGS | 110.02 | 4.86 |
| Our (w/o CUDA) | 6.87 | 1.00 |
| Our | 2.55 | 1.00 |

## 5.2 ABLATION STUDY

Unless otherwise specified, the following experiments are all conducted on the *bicycle* scene of the Mip-NeRF 360 dataset.

**0-1 ILP Superiority in Searching Bit-widths.** In solving the optimal bit-width setting for different attribute channels, we also demonstrate the superiority of the 0-1 ILP. As shown in Tab. 5, we experimented with widely-used General ILP (Yao et al., 2021) and genetic algorithms (Guo et al., 2020; Tang et al., 2024), both of which proved inferior. The 0-1 ILP fully utilizes the size budget

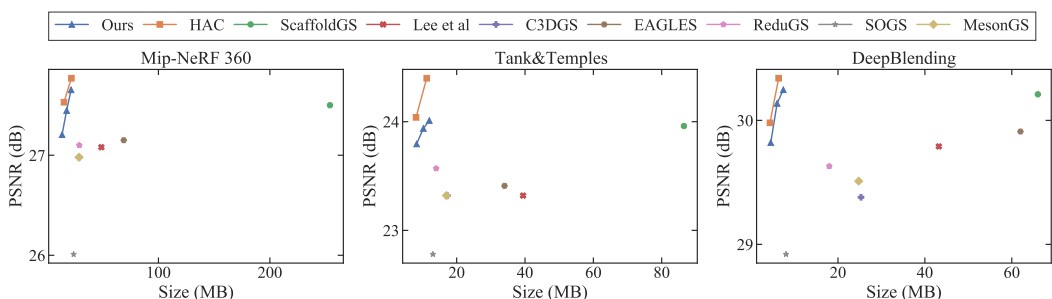

Figure 4: RD curves for quantitative comparisons.

Table 5: 0-1 ILP can search out better results.

| Method | Target size (B) | Searched size (B) | $\Delta$ size (B) | Information loss |
|---|---|---|---|---|
| Genetic algorithm | | 21,833,128 | 8,166,872 | 42,821,038 |
| General ILP | 30000000 | 28,934,805 | 1,065,195 | 1,258,394 |
| 0-1 ILP (Our) | | 29,831,203 | 168,797 | 11,826 |

while minimizing information loss. General ILP involves variables ranging from 1 to 16. In contrast, 0-1 ILP's binary values offer finer control, easier integration of constraints, and more efficient solution techniques. Genetic algorithms, though suited for non-linear or black-box problems, handle constraints less efficiently, making them unsuitable for our linear programming structure.

**Effectiveness of MPQ.** As shown in Fig. 5, we evaluate the effectiveness of each component of our proposed framework. The "baseline" refers to applying MesonGS to ScaffoldGS. We start with the baseline method and progressively incorporate inter-attribute MPQ and intra-attribute MPQ. The results show that the performance is consistently improved with the addition of each module, which proves the effectiveness of hierarchical MPQ.

**Robustness Evaluation**. We evaluated the file size and corresponding performance of the searching algorithm under different numbers of blocks. As shown in Tab. 6, for varying numbers of blocks and different target sizes, our method consistently finds appropriate bit-width settings, ensuring that the final file size is close to the target while maintaining optimal visual quality. Regardless of the block number setting, the final file size and performance are similar, indicating that our method is robust to the number of blocks.

**Why Fixing the Coordinates?** We also explore the necessity of updating coordinates during training. Specifically, we rewrite the backpropagation rules for the voxelization process of the Octree. If, during the forward pass, points within a voxel are deduplicated by averaging, the gradient of that voxel will be evenly distributed to the corresponding points during backpropagation. We find that once backpropagation is enabled for the Octree, the loss fluctuates significantly, ultimately leading to worse fine-tuning results compared to keeping the coordinates fixed.

## 6 CONCLUSION

In this paper, we propose a fine-grained mixed precision compression work to solve the problem of automatically selecting hyper-parameters to compress 3D Gaussians to a target file size while maximizing the visual quality. We propose several key components, including the selection of base model - ScaffoldGS, the selection of compression framework - MesonGS, the size estimator, the inter-attribute mix-precision quantization, and the intra-attribute mix-precision quantization. Extensive experimental results validate the effectiveness of our size-aware 3D Gaussians compression methodology, showcasing a remarkable improvement in the size control of the 3D Gaussian compression and better compression quality compared to SOTA methods. Our method lays the foundation for providing a more controllable solution for the following transmission or streaming tasks.

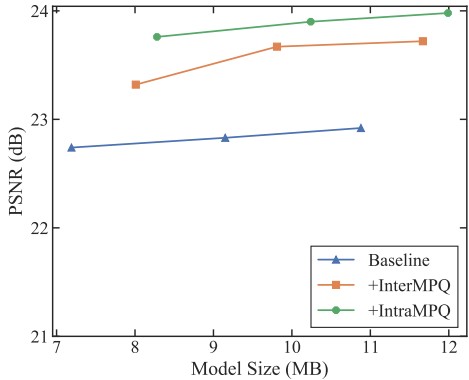

Figure 5: Effectiveness of MPQ.

Table 6: Robustness Evaluation. Mixed-precision quantization can adapt to different $K$, ensuring that the visual quality within a given size is not affected by the $K$.

| N | TgtSize | PSNR | LPIPS | SrchSize |
|---|---------|------|-------|----------|
| 40 |        | 25.16 | 0.2606 | 28.55 |
| 30 | 28.62  | 25.15 | 0.2628 | 29.13 |
| 50 |        | 25.17 | 0.2610 | 29.09 |
| 40 |        | 25.05 | 0.279 | 19.46 |
| 30 | 19.08  | 25.01 | 0.283 | 19.21 |
| 50 |        | 25.08 | 0.278 | 19.59 |

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

# A APPENDIX

Fig. 6 illustrates the difference between HAC and our method. The quantization scheme that we employed is more fine-grained.

Fig. 7, Fig. 8 and Fig. 9 show the loss evolution without anchor, during 0-1000 steps, and during 1000-2000 steps with anchor integrated. It can be observed that the loss increases rapidly after integrating anchor during training, and the overall convergence performance is worse than the baseline training.

Tab. 11, Tab. 12, Tab. 13 and Tab. 14 present the detailed storage composition of our method on MipNeRF 360 dataset, Tank&Temples dataset, Deep Blending dataset, and Synthetic-NeRF dataset.

Tab. 15 summarizes the notations used in this paper and their corresponding definitions.

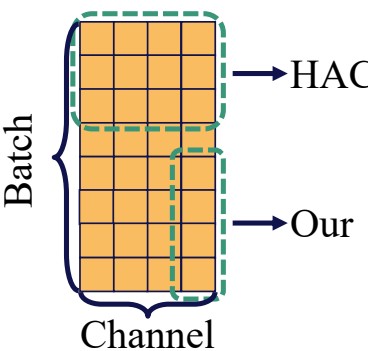

Figure 6: **Parition scheme: HAC vs. Our.**

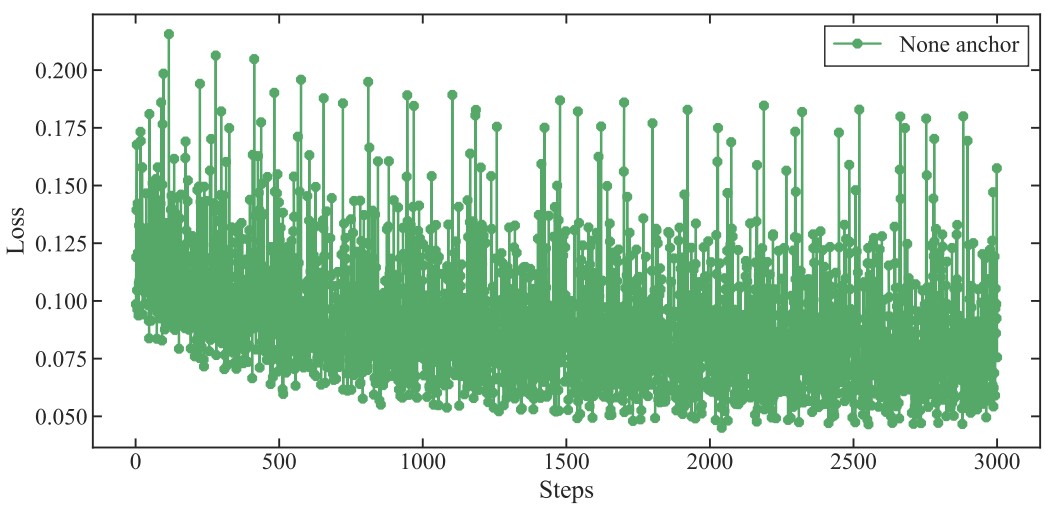

Figure 7: **Loss Evolution without Anchor Integration (Baseline Training)**

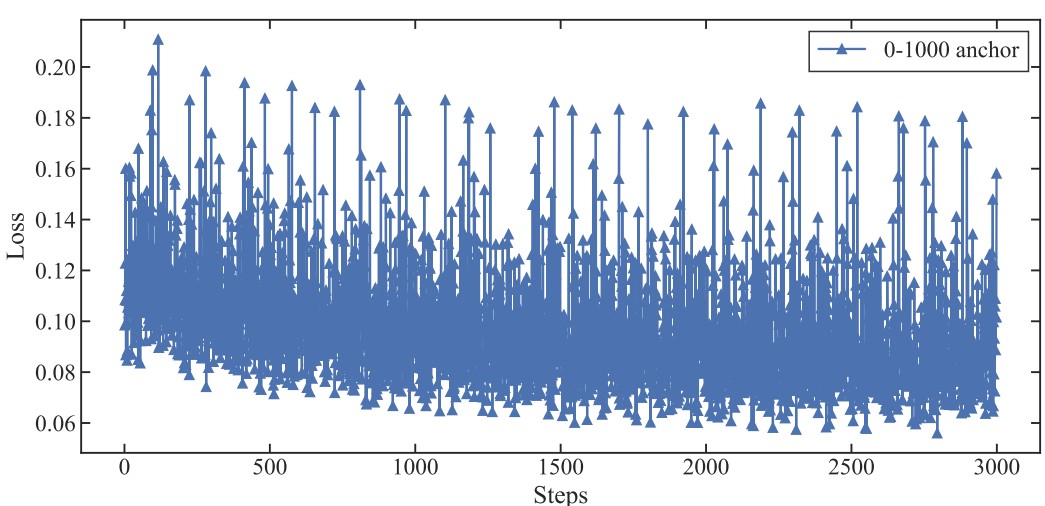

Figure 8: **Loss Evolution during Early Stage (0-1000 steps with Anchor Integration)**

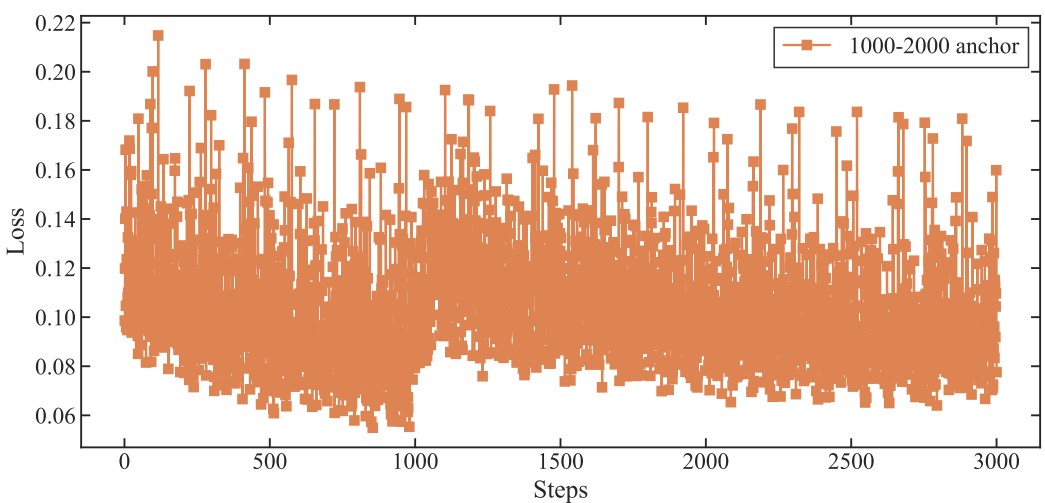

Figure 9: **Loss Evolution during Mid Stage (1000-2000 steps with Anchor Integration)**

Table 7: Per-scene results of Mip-NeRF360 dataset of our approach

| Scene | PSNR | SSIM | LPIPS | Size (MB) |
|---|---|---|---|---|
| *bicycle* | 25.02 | 0.728 | 0.286 | 20.71 |
| *bonsai* | 32.01 | 0.939 | 0.197 | 10.13 |
| *counter* | 28.77 | 0.897 | 0.221 | 6.97 |
| *kitchen* | 30.03 | 0.913 | 0.153 | 7.53 |
| *garden* | 26.46 | 0.813 | 0.186 | 17.90 |
| *room* | 31.32 | 0.917 | 0.217 | 6.63 |
| *stump* | 26.79 | 0.766 | 0.263 | 17.84 |
| *flowers* | 21.30 | 0.577 | 0.379 | 16.62 |
| *treehill* | 23.16 | 0.644 | 0.353 | 17.51 |
| **Average** | **27.21** | **0.799** | **0.251** | **13.54** |
| *bicycle* | 25.16 | 0.745 | 0.261 | 28.56 |
| *bonsai* | 32.47 | 0.944 | 0.189 | 12.48 |
| *counter* | 29.21 | 0.908 | 0.205 | 9.22 |
| *kitchen* | 30.44 | 0.917 | 0.145 | 9.58 |
| *garden* | 26.91 | 0.827 | 0.167 | 23.59 |
| *room* | 31.42 | 0.920 | 0.212 | 7.22 |
| *stump* | 26.86 | 0.766 | 0.262 | 23.28 |
| *flowers* | 21.35 | 0.579 | 0.375 | 22.08 |
| *treehill* | 23.20 | 0.644 | 0.351 | 22.60 |
| **Average** | **27.45** | **0.806** | **0.241** | **17.62** |
| *bicycle* | 25.14 | 0.741 | 0.263 | 35.32 |
| *bonsai* | 32.87 | 0.948 | 0.182 | 16.62 |
| *counter* | 29.45 | 0.913 | 0.195 | 11.39 |
| *kitchen* | 30.87 | 0.923 | 0.136 | 11.43 |
| *garden* | 27.24 | 0.842 | 0.150 | 29.72 |
| *room* | 32.02 | 0.928 | 0.194 | 11.75 |
| *stump* | 26.77 | 0.767 | 0.262 | 29.19 |
| *flowers* | 21.33 | 0.577 | 0.377 | 25.73 |
| *treehill* | 23.20 | 0.644 | 0.352 | 23.49 |
| **Average** | **27.65** | **0.809** | **0.235** | **21.63** |

Table 8: Per-scene results of Tank&Temples dataset of our approach

| Scene | PSNR | SSIM | LPIPS | Size (MB) |
|---|---|---|---|---|
| *truck* | 25.61 | 0.873 | 0.159 | 10.06 |
| *train* | 21.97 | 0.794 | 0.244 | 6.49 |
| **Average** | **23.79** | **0.833** | **0.201** | **8.27** |
| *truck* | 25.79 | 0.878 | 0.152 | 12.47 |
| *train* | 22.08 | 0.792 | 0.243 | 8.01 |
| **Average** | **23.93** | **0.835** | **0.197** | **10.24** |
| *truck* | 25.87 | 0.879 | 0.148 | 14.59 |
| *train* | 22.14 | 0.790 | 0.251 | 9.37 |
| **Average** | **24.01** | **0.835** | **0.199** | **11.98** |

Table 9: Per-scene results of Deep Blending dataset of our approach

| Scene | PSNR | SSIM | LPIPS | Size (MB) |
|---|---|---|---|---|
| *drjonson* | 29.10 | 0.888 | 0.295 | 4.82 |
| *playroom* | 30.54 | 0.904 | 0.278 | 4.24 |
| **Average** | **29.82** | **0.896** | **0.286** | **4.53** |
| *drjonson* | 29.42 | 0.894 | 0.282 | 6.39 |
| *playroom* | 30.85 | 0.908 | 0.268 | 5.59 |
| **Average** | **30.13** | **0.901** | **0.275** | **5.99** |
| *drjonson* | 29.54 | 0.899 | 0.273 | 7.9 |
| *playroom* | 30.95 | 0.907 | 0.269 | 6.93 |
| **Average** | **30.24** | **0.903** | **0.271** | **7.42** |

Table 10: Per-scene results of Synthetic-NeRF dataset of our approach.

| Scene | PSNR | SSIM | LPIPS | Size (MB) |
|---|---|---|---|---|
| *chair* | 33.53 | 0.979 | 0.020 | 1.18 |
| *drumps* | 25.81 | 0.945 | 0.049 | 1.73 |
| *ficus* | 34.29 | 0.982 | 0.016 | 1.09 |
| *hotdogs* | 36.67 | 0.979 | 0.030 | 0.73 |
| *lego* | 33.93 | 0.972 | 0.030 | 1.39 |
| *materials* | 29.94 | 0.956 | 0.046 | 1.65 |
| *mic* | 35.44 | 0.989 | 0.010 | 0.89 |
| *ship* | 30.90 | 0.896 | 0.127 | 1.78 |
| **Average** | **32.56** | **0.962** | **0.041** | **1.30** |
| *chair* | 34.10 | 0.981 | 0.017 | 1.58 |
| *drumps* | 25.96 | 0.947 | 0.047 | 2.32 |
| *ficus* | 34.64 | 0.983 | 0.015 | 1.45 |
| *hotdogs* | 37.03 | 0.981 | 0.027 | 0.97 |
| *lego* | 34.20 | 0.975 | 0.026 | 1.80 |
| *materials* | 30.21 | 0.958 | 0.044 | 2.13 |
| *mic* | 35.97 | 0.990 | 0.009 | 1.18 |
| *ship* | 31.10 | 0.898 | 0.123 | 2.39 |
| **Average** | **32.90** | **0.964** | **0.038** | **1.73** |
| *chair* | 34.71 | 0.983 | 0.014 | 1.92 |
| *drumps* | 26.05 | 0.947 | 0.047 | 2.82 |
| *ficus* | 34.86 | 0.984 | 0.014 | 1.78 |
| *hotdogs* | 37.49 | 0.982 | 0.025 | 1.17 |
| *lego* | 35.08 | 0.978 | 0.021 | 2.24 |
| *materials* | 30.41 | 0.959 | 0.042 | 2.69 |
| *mic* | 36.39 | 0.991 | 0.008 | 1.42 |
| *ship* | 31.29 | 0.900 | 0.118 | 2.89 |
| **Average** | **33.28** | **0.965** | **0.036** | **2.12** |

Table 11: Composition of Storage Sizes (MB) for Different Parts in the Mip-NeRF360 Dataset

| Scene | Octree | Metadata+Attributes | MLP |
|---|---|---|---|
| *bicycle* | 1.360 | 27.19 | 0.048 |
| *bonsai* | 0.717 | 12.65 | 0.048 |
| *counter* | 0.491 | 8.69 | 0.048 |
| *garden* | 1.190 | 22.66 | 0.048 |
| *kitchen* | 0.331 | 9.21 | 0.048 |
| *room* | 0.431 | 9.01 | 0.048 |
| *stump* | 1.010 | 22.22 | 0.048 |
| *treehill* | 0.917 | 21.65 | 0.046 |
| *flowers* | 0.941 | 20.89 | 0.046 |

Table 12: Composition of Storage Sizes (MB) for Different Parts in the Tank&Temples Dataset

| Scene | Octree | Metadata+Attributes | MLP |
|---|---|---|---|
| *train* | 0.243 | 6.21 | 0.046 |
| *truck* | 0.402 | 9.62 | 0.046 |

Table 13: Composition of Storage Sizes (MB) for Different Parts in the Deep Blending Dataset

| Scene | Octree | Metadata+Attributes | MLP |
|---|---|---|---|
| *drjohnson* | 0.269 | 6.08 | 0.048 |
| *playroom* | 0.245 | 5.31 | 0.048 |

Table 14: Composition of Storage Sizes (MB) for Different Parts in the Synthetic-NeRF Dataset

| Scene | Octree | Metadata+Attributes | MLP |
|---|---|---|---|
| *chair* | 0.073 | 1.47 | 0.046 |
| *drums* | 0.098 | 2.19 | 0.046 |
| *ficus* | 0.151 | 1.36 | 0.046 |
| *hotdog* | 0.101 | 0.89 | 0.046 |
| *lego* | 0.187 | 1.67 | 0.046 |
| *materials* | 0.213 | 1.99 | 0.046 |
| *mic* | 0.118 | 1.09 | 0.046 |
| *ship* | 0.248 | 2.25 | 0.046 |

Table 15: Notation table.

| Notation | Definition |
|---|---|
| $\Sigma$ | Covariance matrix of a Gaussian distribution |
| $G(x)$ | Gaussian function |
| $\mathbf{S}$ | Scaling matrix |
| $s$ | Scaling vector |
| $\mathbf{R}$ | Rotation matrix. |
| $\mathbf{q}$ | Rotation quaternion |
| $\mathbf{W}$ | Viewing transform |
| $\mathbf{J}$ | Jacobian matrix |
| $\mathbf{\Sigma}'$ | Covariance matrix of in camera space |
| $\mathbf{SH}$ | Spherical harmonics co-efficient |
| $o$ | Opacity |
| $\mu$ | 3D center of a Gaussian function |
| $\mathbf{x}$ | Location of anchor point in ScaffoldGS |
| $\mathbf{f}$ | A context feature of the anchor point |
| $\mathbf{l}$ | Scaling factor of ScaffoldGS |
| $\mathbf{O}$ | Learnable offsets of ScaffoldGS |
| $\mathcal{A}$ | Attributes of ScaffoldGS, including features $\mathbf{f}$, scaling factor $\mathbf{l}$, and learnable offsets $\mathbf{O}$. |
| $\tau$ | Pruning percentage |
| $N$ | Number of anchor points after applying pruning and voxelization |
| $H$ | Number of channels of ScaffoldGS's attributes |
| $d$ | Depth of octree |
| $K$ | Number of blocks |
| $B$ | Quantization options, usually set as 16 |
| $Q$ | One-hot bit precision matrix |
| $\mathbf{w}$ | Weight vector, to transform $Q$ into bit vector |
| $\Omega$ | Information loss matrix for integer linear programming |
| $S(\cdot)$ | Function of size estimator |
| $b_i$ | A block of attribute |
| $\hat{b}_i$ | The vector that de-quantized from the quantized $b_i$ |
| $n_i$ | The Start index of the block $b_i$ |
| $F(k, l)$ | The minimal total information loss when slicing the $l$ elements into $k$ blocks |
| $U$ | Step unit of dynamic programming |
| $L(n_i, n_{i+1})$ | Information loss in quantizing block $b_i$ |

