# OpenReview forum: "Size-aware Compression of 3D Gaussians with Fine-grained Mixed Precision Quantization"
_ICLR.cc/2025/Conference — ICLR 2025 Conference Withdrawn Submission_

### Official Review · Reviewer_EpWC · 2024-10-29

**Soundness:** 3
**Presentation:** 2
**Contribution:** 2
**Rating:** 5
**Confidence:** 3

**Summary:**

In this paper, the authors propose a mixed-precision quantization method for 3DGS compression. Specifically, different bit-widths are assigned to different attribute channels of the gaussians. In addition, each attribute channel is partitioned into blocks of vectors. While previous methods require completing the entire compression process to determine the compressed file size, the proposed method introduces a size estimator to determine the model size within 70 seconds. Experiments show that the proposed method improves the performance of fine-tuning-free approaches.

**Strengths:**

(1) The motivation is clear and this paper is easy to follow.
(2) Superior performance as compared with previous methods.

**Weaknesses:**

(1) My major concern is about the marginal performance gain. In Table 2, it seems the proposed method is even inferior to HAC, especially on the Tanks&Temples dataset. As compared to HAC, our-small has similar model size but produces lower performance on all metrics. For our-large, superior performance is acheived at the cost of much larger model size. So I wonder the superiority of the proposed method as compared to HAC. I can see that the proposed method is finetuning-free, but the authors should clarify which methods are fair results as compared to the proposed method.
(2) Following the first comment, as shown in Fig. 4, the PSNR score of the proposed method seems to be lower than that of HAC under the same size. This further shows that the proposed method does not produces either higher accuracy or better efficiency. So the effectiveness of the proposed method seems to be further highlighted.
(3) As mix-precision quantization is one of the major contributions for the proposed method, the bit-widths for different attribute channels should be discussed, which could be an interesting point for follow-up researchers. It would be better if the  bit-widths for different attribute channels under different buget can be further analyzed.
(4) Typos:
- Line 40: 5.2710^6?
- MPQ is not defined in the paper

**Questions:**

Please see weaknesses.

---

### Official Review · Reviewer_3Yz8 · 2024-10-31

**Soundness:** 3
**Presentation:** 3
**Contribution:** 3
**Rating:** 6
**Confidence:** 3

**Summary:**

This paper introduces a method for compressing 3D Gaussian models to meet target file sizes while preserving visual quality. The key contributions include a quick size estimator for compression prediction, a hierarchical mixed precision quantization approach using integer linear programming and dynamic programming, and a complete compression pipeline that finds optimal parameters 100x faster than existing methods. The approach is validated on standard datasets, showing competitive results in both compression speed and visual quality metrics.

**Strengths:**

- Formulation of the compression problem that explicitly considers target file sizes, addressing a practical need not well-handled by existing methods
- Combination of size estimation with mixed precision quantization, offering a new approach to balancing compression and quality
- The original use of 0-1 ILP for bit-width selection in 3D Gaussian compression, adapting techniques from neural network quantization to a new domain
- Clear justification for design choices (e.g., choosing MesonGS over other frameworks due to size stability)
- 100× speedup in parameter search makes the method much more practical for real-world applications

**Weaknesses:**

- I'm concerned about the paper's core assumption that attribute channels are independent during quantization. This feels like a significant oversimplification without any supporting evidence. I would like to see some experiments showing whether there's a correlation between channels' quantization errors and how this impacts the final results.
- Why only test on static scenes with a single file size target (30MB)? For a paper claiming to be "size-aware," I'd expect to see results across various target sizes and more challenging scenarios like dynamic scenes. I'm particularly curious how their method handles SH coefficients under different lighting conditions.
- The performance analysis feels incomplete. We get plenty of quality metrics, but what about memory usage during compression? Also, they mention using CUDA for speed-up but don't explain the implementation details - this kind of information is crucial for anyone trying to replicate their work.
- The paper shows how their method works but doesn't really explain how to use it. How do we choose the step size U or the number of blocks K in practice? Table 6 shows it's robust to different K values, but I'm still wondering what values I should pick for my use case.
- I'm worried about error propagation in their system. What happens when errors from the inter-attribute stage combine with those from the intra-attribute stage? And how does the method behave with very small target sizes? Some analysis of failure cases would really help understand the method's limitations.

**Questions:**

- Could you provide more details about your method's performance on dynamic scenes, particularly regarding temporal coherence and compression consistency between frames?
- What are the memory-speed trade-offs in your compression pipeline, and how does the peak memory usage compare to existing methods?
- Have you identified any quality cliffs or failure cases where the compression performance degrades significantly (e.g., minimum achievable file size, complex geometries, or detailed textures)?

---

### Official Review · Reviewer_JTqY · 2024-11-04

**Soundness:** 3
**Presentation:** 3
**Contribution:** 3
**Rating:** 6
**Confidence:** 3

**Summary:**

The paper presents a novel approach to size-aware compression of 3D Gaussians, focusing on fine-grained mixed precision quantization to optimize file size while maximizing visual quality. The authors propose a framework that includes several key components: the selection of a base model (ScaffoldGS), a compression framework (MesonGS), and a size estimator.

**Strengths:**

(1) This is a well-written paper.
(2) The proposed method is compared with various methods. The experiments are complete and convincing
(3) Some visualizations are helpful to understand.

**Weaknesses:**

（1）Lack of FPS comparisons

**Questions:**

（1）In line 202, how do you obtain the average important score of anchors in detail?

---

### Official Review · Reviewer_U8m2 · 2024-11-04

**Soundness:** 2
**Presentation:** 2
**Contribution:** 1
**Rating:** 3
**Confidence:** 4

**Summary:**

This paper presents a size-aware 3DGS compression approach designed to achieve accurate size estimation. Building upon the MesonGS framework, the authors first develop a size estimator to obtain precise size measurements. To enhance performance further, a mixed-precision quantization strategy that incorporates 0-1 integer linear programming and dynamic programming is proposed. Experimental results demonstrate that the proposed method achieves superior compression quality compared to existing approaches while requiring less search time.

**Strengths:**

1. The proposed method significantly reduces search time relative to existing approaches while ensuring accurate size estimation and strong compression performance.

**Weaknesses:**

1. The novelty of this paper is limited. The overall coding architecture closely resembles that of MesonGS, with only marginal innovations. The primary contributions consist primarily of technical enhancements, specifically 0-1 integer linear programming and dynamic programming, rather than presenting novel research insights.
2. The application of the proposed method is confined to MesonGS, which restricts its potential use cases. To demonstrate the effectiveness of the method, it would be beneficial to apply it to multiple baseline models.
3. The performance gains attributed to the proposed method are not adequately analyzed. Given that the core idea and methodology focus on accurate size estimation, the substantial performance improvement over MesonGS (as shown in Table 2, with Mip-NeRF 360 increasing from 26.68 dB to 27.65 dB) appears insufficiently justified. A detailed analysis of the contribution of each component, including the transition from 3DGS to Scaffold-GS, the proposed mixed-precision quantization strategy, and the fine-tuning process, is warranted.
4. There are several writing issues. For example, “PNSR” in Table 1 should be “PSNR”. Additionally, notations should be defined upon their initial appearance, such as “Ai” in Equation (4) and the “⊙” symbol in Equation (5).

**Questions:**

Please see weaknesses.

---

### Note · Authors · 2024-11-12

**Comment:**

Thank you to all reviewers and the Area Chair for your thoughtful and detailed feedback on our submission. We are very grateful for the time and effort each of you has dedicated to evaluating our work. Your insights have provided us with valuable directions to improve our research. After careful consideration, we have decided to withdraw the paper in order to address these suggestions more comprehensively. We look forward to using this feedback to refine our work and hope to submit an improved version in the future. Thank you again for your invaluable support and constructive input.

**Withdrawal Confirmation:**

I have read and agree with the venue's withdrawal policy on behalf of myself and my co-authors.